# Mental Health and Aggression in Indonesian Women

**DOI:** 10.3390/bs13090727

**Published:** 2023-08-31

**Authors:** Aryati Hamzy, Cheng-Chung Chen, Kuan-Ying Hsieh

**Affiliations:** 1Department of International Graduate Program of Education and Human Development, College of Social Sciences, National Sun Yat-sen University, No. 70 Lianhai Rd., Gushan District, Kaohsiung City 804, Taiwan; 2Department of Post-Baccalaureate Medicine, College of Medicine, National Sun Yat-Sen University, Kaohsiung City 804, Taiwan; 3Department of Child and Adolescent Psychiatry, Kaohsiung Municipal Kai-Syuan Psychiatric Hospital, Kaohsiung City 802, Taiwan; 4Department of Physical Therapy, I-Shou University, Kaohsiung City 824, Taiwan

**Keywords:** adults, aggression, Indonesian women, mental health, multicultural, stigma

## Abstract

Aggression is a global problem and complex social behavior. In Indonesia, some common manifestations of aggression are sexual harassment, domestic violence, and the stigmatization of other people. However, unlike men, aggression in women is still rarely studied, whereas facts find that many conditions can make women more vulnerable. There are various aspects related to biological, psychological, social, and cultural issues that can potentially provoke female aggression. Furthermore, mental health and aggression are often viewed as an automatic association and are inseparable in society, reinforcing the stigma against people with mental problems, particularly women, who tend to suffer more stigma of mental health issues than men. However, there has not yet been a study explicitly related to this relationship in the general population of women. The current study aims to examine the overall relationship between mental health and aggression in the extensive general population of Indonesian women with various mental conditions ranging from a normal mental state to severe mental health problems. This was a cross-sectional study conducted using uncontrolled quota sampling via distributing online self-report questionnaires, the modified Indonesian version instruments of the Brief Symptoms Rating Scale-5, and the Buss Aggression Scale with high internal consistency. This study among 203 women aged 19–67 in Indonesia, a multicultural nation and the fourth densest country in the world, proposes that mental health can be a predictor of aggressive behavior, with the influence of mental health on the aggression of women being 21.6% only. The finding indicates that mental health issues are not a macro contributing factor to women’s aggressiveness in society and may help reduce stigma against women with mental health problems.

## 1. Introduction

Women suffer from discrimination and violence all over the world, yet they are underreported. Their participation in economic and political decision making is also underrepresented [1]. Furthermore, the number of women worldwide who are physically, sexually, or psychologically abused during their lifetime is one out of three [2]. The same thing happens in Indonesia, despite state laws regulating discrimination and violence against women; they still occur frequently and the prevalence of sexual harassment and domestic violence remains high yearly [3]. The phenomenon of mental health problems in Indonesia is still a thorny issue, including the stigma attached to people with mental health problems [4]. A study by Vázquez et al. (2021) among homeless people found that the percentage of women who face violence is higher than that of men. Additionally, women face higher obstacles in finding jobs and have a higher tendency to become homeless more than five times [5]. During the COVID-19 pandemic, the risks of experiencing violence, discrimination, and poor mental condition tended to increase with a significant association among the three variables [6]. Global disparities in women’s mental health and well-being are profoundly influenced by the dominancy or power differentials between men and women, access to resources in society and at home, and sexual and domestic violence, according to the World Health Organization reports [7]. Increased mental health problems and a lack of well-being in women, mainly among girls and young women, are global trends in recent times [8]. 

To explore mental health care for women, it is important to note how women’s mental health challenges are different from those of men. Studies suggest that women are more vulnerable to mental health problems such as depression, largely because women’s brains differ from those of men [9,10]. A person’s mental health issues are usually caused by an interaction of biological factors, such as chemical imbalances of neurotransmitters and steroid hormones in the brain, genes, brain diseases or injury, and psychosocial factors, such as their roles and experiences in society [9,11]. Women’s roles are challenging; they tend to be the primary caregivers for their children, aging parents, and spouses in the family, and at the same time, have similar paths of work and education as men, even though the gender issue might affect psychology and behavior differently across cultures [12]. All experiences of discrimination, violence, poverty, and the hassle of daily duties in home life can potentially cause stress and impact their health both psychologically and physically [7,13], which further influences their psychological well-being and behavior, including aggressive behavior. 

Montoya et al. (2012) stated that the role of the steroid hormone testosterone, the stress hormone cortisol, and the neurotransmitter serotonin are key regulators of human social aggression [14]. However, a finding by Shors and Leuner (2003) indicates that women and men can react to the same emotional event using different hormonal and neural mechanisms, wherein the response of women to stressful experiences is unique, dependent on the changing levels of the steroid hormone estrogen [10]. Other studies also found the influence of estrogen levels on women’s mental health [15,16] and the impact of estrogen on serotonin degradation inhibition [15]. A study by Cashdan (2003) states that high testosterone and low estrogen levels contribute to female competitiveness [17].

Aggression is a complex social behavior and one of the social problems in the world [18,19], including in Indonesia, the fourth densest nation on earth [20], with a multicultural society where each island has different ethnicities, traditions, and values [21]. Aggressive behavior in various forms including physical, verbal, active, passive, direct or indirect in nature, aim to harm a person’s reputation, and is a global phenomenon in both men and women [18]. A cross-cultural study by Burbank (1987) across 317 societies found that women’s aggression is primarily indirect and seldom physical as they fear being harmed, for example, spreading false rumors, gossiping, excluding others from social groups, insinuating without directly accusing, and criticizing the looks or personalities of others [22,23]. The victims will possibly experience repeated abuse from the offenders, and the experience of victimization, rejection, neglect, or suffering will increase the risk of having aggressive and antisocial behavior in the future [24]. Studies in Indonesia found that bullying, both verbally and non-verbally, as one of the aggression forms, is an everyday thing that occurs in schools [25]. However, violence and aggression are usually viewed as male issues, whereas women’s aggression is still an understudied comparison to men’s aggression, which cannot be overlooked. Therefore, it is necessary to conduct studies to identify how to deal with aggression problems, particularly in the female population, including in Indonesian women. 

In relation to mental health issues, most people may perceive aggression as an automatic association with mental health issues, regardless of how much influence mental health has on aggression, reinforcing the stigma against people with mental health problems. In society, violence and mental disorders are often viewed as inseparable, causing harsh stigmas for individuals and their families [26], and even sometimes for professionals and institutions who deal with them. Studies suggest that dangerous behaviors, uncommonness, and emotional deviations are primary causes of mental disorder labels or stigma [27,28]. Compared to men, women usually suffer more stigma from mental health issues because they tend to internalize stigma and society’s perceptions with all of the consequences [29,30]. A study by Drury et al. (2023) reveals that both self-stigma and public stigma reduce the willingness of individuals to seek help [31]. However, in the general population of women in society, there has not yet been a study explicitly related to the overall association between mental health and aggression with various mental conditions ranging from normal mental state to severe mental health problems; more specifically, there is still no valid data on what percentage of this mental health issue can influence women’s aggressiveness in the community, even though the impact of mental disorders such as mania, substance use disorders, and schizophrenia, on violence and aggression in psychiatric settings is commonly observed. Findings suggest that up to fifty percent of psychiatric disorder patients exhibit aggression, compared with less than two percent of the general population. In fact, most of society’s aggression generally results from people who do not have mental disorders [32]. Therefore, this study aims to examine the broader relationship between mental health and aggression in the general population of females, in this case, of the Indonesian women adult population with various mental conditions, including a normal mental state. 

## 2. Materials and Methods

### 2.1. Participants

Using an online survey approach, this cross-sectional study used an observational analytical design and included women aged 19 years and older. Uncontrolled quota sampling, a sampling method that does not rely on probability, was used to select participants based on the percentage of the Indonesian population on the two major islands and some minor islands in Indonesia, where all population on those islands cover more than 60% of the total Indonesian population in multicultural contexts. To ensure samples could be represented well, the researchers divided Indonesian territory into two areas, West Indonesia and East Indonesia, and each Indonesian area consists of one major island and several minor islands, wherein the selection criterion of a major island in each area was determined by the highest population percentage. All participants were captured from all provinces of each major island and several minor islands/provinces according to the percentage of their populations.

### 2.2. Instruments

Self-reported online questionnaires were composed of two Indonesian versions modified for online use, was validated and proven reliable among a valid sample population and showed a high level of internal consistency. The Brief Symptoms Rating Scale (validity index = 0.786–0.854; Cronbach’s alpha = 0.881) consists of 5 items and each item is scored using a five-point Likert scale in which the choices range from “not at all” to “extremely”. All items are favorable items (0 = not at all; 1 = a little bit; 2 = moderately; 3 = often; 4 = extremely). This instrument of Brief Symptoms Rating Scale-5 (BSRS-5) was developed from the Brief Symptoms Rating Scale-50 (BSRS-50) by Lee et al. (2003) to identify common psychological distress [33], wherein the BSRS-50 modification was carried out by Lee et al. (1990) from the Symptom Checklist-90 R (SCL-90-R) [34]. They assessed BSRS-5 reliability and validity in a variety of populations, including the general population and psychiatric medical settings, and all of the results showed that the BSRS-5 performed well. The Buss Aggression Scale (validity index = 0.351–0.702; Cronbach’s alpha = 0.914) consists of 28 items for adults based on Buss’s theory of aggressive behavior [18]. The questionnaire asked about aggression aspects that were physical or verbal, directly or indirectly, and actively or passively. The form of aggression was expressed as negative aggression toward living things or tangible objects. There are four choices ranging from “always” to “never” for each item based on the detailed frequency criteria in the last three months or one month. All items are favorable items (4 = always; 3 = often; 2 = seldom; 1 = never). 

### 2.3. Procedures

By sharing the online self-report questionnaires via WhatsApp on mobile phones with potential participants and community groups, the researcher gathered data via an internet survey using Google Forms. The use of the WhatsApp application on smartphones, which is favored and has been widely used in both urban and rural communities in Indonesia, can effectively capture the potential sample both individually and in groups via dissemination assistance from the social networks of professional organizations, colleagues, community leaders, and others spread across each island and province in Indonesia. The Google Forms that contain questionnaires were preceded by a message notification to stress the legitimacy and importance of the online survey. Data from this study, which used only women’s samples, were part of the main data research. The research was approved by the Medical and Health Research Ethics Committee of Gadjah Mada University (protocol number KE/0148/02/2022 and date of approval 7 April 2022) and all procedures followed the Helsinki Declaration guidelines and its later amendments. Informed consent was obtained from all the participants. 

### 2.4. Data Analysis

IBM SPSS Statistics 25 was used to analyze the data, which was preceded by examining data using descriptive statistics to determine whether the range of values was plausible. Then, the researcher continued to verify if both continuous variables met all assumptions of simple linear regression analysis, which were normality, linearity, no significant outliers, and having the independence of observations. This simple linear regression analysis was performed to identify the association between mental health and aggression in Indonesian women.

## 3. Results 

There were 206 Indonesian women from different islands and different subcultural environments who participated in this study, but only 203 cases of samples fulfilled the assumptions of simple linear regression analysis. The descriptive information showed the three extreme outliers and a substantial positive skewness on the aggression variable only. According to Tabachnick and Fidell (2001), if the cases of outliers represent 5% or less of the total sample size, the deletion of outliers becomes a viable option [35]. After deleting the three extreme outliers on the dependent variable and continuing with data transformation using square root transformation to improve the normality of variables [36], the distribution of residuals became normal, and all assumptions were fulfilled. 

### 3.1. Descriptive Analysis

The descriptive analysis results of the study that examined the relationship between mental health and aggression in the general women population in Indonesia can be seen in Table 1.

Table 1 represents the 203 participants, all of whom were women aged 19–67 years old with a mean age (SD) of 36.7 (12.9) years old. They consisted of 119 young adult women aged 19–40 years old (58.6%), 83 women in middle years aged 41–65 years old (40.9%), and 1 elderly woman aged > 65 years old (0.5%). Most of them grew up and live in urban areas (65%). The mental health score ranged from 0 to 20, with a mean score (SD) of 6.6 (5.0). The aggression behavior score ranged from 28 to 56, with a mean score (SD) of 37.0 (6.8). There were 103 participants who had normal mental health conditions (50.7%), 49 participants with mild mental health problems (24.1%), 34 participants with moderate mental health problems (16.8%), and 17 participants with severe mental health problems (8.4%). The total percentage of moderate and severe problems in the women young adult group (33.6%) was higher than in the women middle years group (13.3%), but the total percentage of the normal condition and mild problems in the women young adults’ group (66.4%) was lower than in the women middle years group (86.7%).

### 3.2. Variable Analysis

The summary of the statistical analysis results between both variables, mental health and aggression in the general Indonesian women population, can be seen in Table 2.

Table 2 shows that the results of the study proposed a highly statistically significant positive relationship between mental health and aggression in the adult women population in Indonesia (r = 0.465, *p* < 0.001) with an α level ≤ 0.05. Therefore, it could be stated that the null hypothesis was rejected. The higher the mental health problems score, the higher the aggression score. It found a moderate positive correlation between mental health and aggression, with the coefficient of determination being 21.6% (r-squared = 0.216). This finding suggests that mental health affected 21.6% of women’s aggression, whereas the remaining 78.4% was predisposed by other elements.

According to Table 2, it shows that in Indonesian women both young adults and middle-aged, there were statistically significant positive relationships between the two variables, and the correlations were moderately positive correlations for both the women young adult group (r = 0.487, *p* < 0.001) and the women middle-year group (r = 0.350, *p* = 0.001). They attained the 5% significance level.

The graph of the relationship between mental health and aggression in the general population of Indonesian women in Indonesia, can be seen in Figure 1.

The scatterplot displays an uphill linear line pattern with a moderate and positive direction of correlation between two quantitative variables, mental health and aggression. The statistical analysis fulfilled the assumption that the relationship between the two variables was linear, and its result attained the 5% significance level. Therefore, it can be concluded that there was a significant positive linear relationship between mental health and aggression in the female population.

The graph in Figure 2 shows two lines with an uphill linear pattern. The continuous line belonging to the women’s young adult group was longer and had sharper steepness than the dotted line belonging to the women’s middle-year group. This means that the correlation between the two variables in the women young adults’ group was stronger than in the women middle years group, and there were no samples in the women middle years group that reflected the cases of severe mental health problems that scored from 15 to 20. However, both age groups had significant positive linear relationships between mental health and aggression in the women population.

## 4. Discussion

### 4.1. All Participants

Indonesia is one of the most prominent nations, although Indonesia is still a developing country due to its position as the largest archipelagic country in the world, with 17,500 islands and is the fourth densest nation on earth after China, India, and the United States. It is also the largest country in Southeast Asia because of its distance, an eighth of the earth’s circle. As a unique and highly multicultural society, Indonesia consists of around 1340 ethnic groups and 719 languages [20,37]. All of these facts make Indonesian women capable of representing the research population of women internationally.

The number of participants in the young adult group was the highest compared with the middle-year and later-year groups. However, most of the overall participants had a low level of aggression (99.5%) and the remaining 0.5% was a moderate aggression level case only. It is beyond reality that this is a lower picture than expected. This result is possible because the participation of samples in this study tends to be voluntary, so it allowed most participants who tend to have pro-social behavior, i.e., like to help others, to engage in this study, besides which, the situation of the COVID-19 pandemic can hinder the prospective participants with higher aggression level to participate. A longitudinal study by Eron and Huesmann (1984) revealed that there is a negative correlation between pro-social behavior and anti-social aggression [38]. Nonetheless, if the number of samples increases, it may obtain more participants with higher aggression levels because of the desire to know their mental health condition, and the relatively short survey can encourage individuals with higher aggression to participate. This study, which used the simple linear regression analysis, found that the influence of mental health problems on women’s aggression in the general community was only 21.6%, indicating mental health issue contributes a minor amount in women’s aggressiveness, as well as distinct predictors may have a bigger role in the aggression of females. A study by Thummanond and Maneesri (2020) in Thailand found that self-control had a 76.0% influence on women’s aggression [39], which means self-control may be a more significant predictor of aggressiveness than mental health status in Indonesian women. However, it is assumed that in non-provocation conditions, the percentage of the mental health influence on the aggressiveness of men may be higher than in women because men’s emotional response differs from women, who display higher aggression in high-stress situations, while women indicate a lower aggression intensity level in facing the same stress condition but increasing their level of sadness [40]. 

According to an effective screening instrument using the self-reported BSRS-5 for assessing common psychological distress, the number of participants with mental health problems ranging from slight to severe was almost half of the total participants (49.3%). This high percentage may represent the percentage of mental health problems in the population of women in Indonesia, which may be influenced by the COVID-19 pandemic. However, the total number of participants with moderate and severe mental health problems, where participants should seek psychological counseling and medical services, was 25.2%, which may represent a prevalence of mental disorders in the Indonesian women population during the pandemic. The rest were slight problems (24.1%), where there was no need to seek mental health professionals, and normal mental conditions (50.7%). The study results were in line with the information from the Ministry of Health in 2021 that the prevalence of individuals with mental disorders in the Republic of Indonesia was regarding 20% of the overall population [4], wherein the percentage continues to increase during the COVID-19 pandemic. An online survey conducted by Rakuten Insight in 2022 found that 57% of the 9411 participants aged 16 years and older in Indonesia had mental health problems. This percentage of 57% exceeding the 49.3% in this current study may be due to the age range being more expansive, the survey time & survey tool difference, and the larger sample size [41].

Individuals with mental health problems are vulnerable to mistreatment by others and may have a higher level of victimization because of the impact of decreasing self-efficacy and coping mechanisms [42]. Studies show that the roles of the victim and offender often overlap, where most offenders with anti-social aggression have been victims, even though aggression victims do not always end up as offenders [43]. As a result of victimization itself, women are prone to mental health problems, and some internalize the victim role developing a victim mentality, resulting in emotional suffering and destructive behavior [44]. Therefore, we can presume that the internalization of the victim role (victim mentality) among women from conditions such as discrimination, neglect, rejection, and sexual and domestic violence may influence their mental health and reinforce the occurrence of anti-social aggression in their daily lives, thus increasing the aggression level. In other words, the higher the victim mentality, the higher the influence of mental health problems on aggression. So, researchers assume that a victim mentality can moderate the relationship between the two variables. However, Karremans and Van Lange (2004) stated that a high moral identity in the form of the ability to forgive and empathy and altruism can prevent people who experience victimization from becoming perpetrators of aggression and resolve conflicts [45].

### 4.2. The Age of Young Adults vs. Middle Years

In this study, the correlations between both variables were stronger in the young adult women group than in the middle-year women group, likely because middle-year women tend to have more agreeable abilities, better self-regulation, and more maturity in self-control than young adult women [46]. A study among young adults of 67 college students by Shorey et al. (2015) reported that poor self-regulation moderates the relationship between negative affect (such as depression, anger, and hostility) and the violence of intimate partners [47]. In addition to being potential practices across a broad spectrum of human behavior, another study also found that self-control can be used to predict and reduce aggression [48]. The results of the study also found that the percentage of Indonesian women with mental health problems was higher in young adults (56.3%) than in the middle years (39.8%). These results are consistent with the finding that while mental health problems are common and one of the most common problems in young adults, they are less likely to seek and receive help [8]. The reasons why it occurred may be because young adulthood is a steady search period and time reproductive, which is a period full of problems and emotional tension, a period of commitment, and adjustment to a new lifestyle [49].

During the young adult period, most Indonesian women deal with pregnancy and childbirth with all of their impacts, including the change in hormonal levels, in addition to acting as a wife, worker, and other social roles in the extended family and society. Furthermore, the interaction of this biological factor and psychosocial stressors in life can influence mental health conditions [11,50]. A study by Lee and Powers (2002) proposed that young women who have multiple roles with four roles or more are very vulnerable to physical and mental health problems [51]. Problems of violence, mainly domestic violence and sexual harassment, are commonly faced by Indonesian women as well, especially young people, which makes things worse [3]. Zarbaliyev (2017) states that different subcultural contexts of the multicultural environment in Indonesia, such as ethnicities, traditions, values, religions, and social status, may make a large contribution to conflicts, violence, and aggression problems if there is no great tolerance for each other in the community [21]. In addition, this cultural diversity can have an impact on psychological outcomes such as enriched cognition and behavior, as well as frustration when it leads to cultural mixing [52].

In the middle years, generally, humans reach the peak of their achievements. Nevertheless, it is also a time of stress due to various physical changes that can have negative psychological impacts. In a middle-year group of women, menopausal symptoms caused by changes in sex hormone levels, mainly estrogen, also have negative impacts on both physical and psychological health [16]. However, they may be in a steadier condition and have better psychological adjustments because of a more mature age via various life experiences, and the attention to spirituality is greater than that of young adults, supporting them when coping with stress [53]. The study did not have samples with severe mental health problems in the middle-year women group; however, if the sample size of middle-year women is increased, it is possible to obtain severe mental health problem cases in this group, but perhaps the case number will be less than that in the young adult women group.

### 4.3. Limitations and Implications

The limitation of the study is the issue of a larger sample size, so that each sample group can be represented well, specifically in the age group of later years and the higher aggression levels group. The non-randomized sampling method allowed the participants in this study to voluntary join, and the COVID-19 pandemic situation can hinder and reduce the motivation of the prospective participants with higher aggression levels to participate, so it was more challenging to find the samples with levels of moderate and high aggression. However, these limitation issues have no macro influence on the research value as long as the appropriate data analysis method is selected. Despite the limitations, the study still has important implications for overcoming aggression problems via prevention and intervention programs for women’s mental health in society, particularly in the young adult women group. These findings can encourage greater attention to mental health, specifically in women, one of the most neglected issues worldwide [50]. Moreover, it may decrease the stigma against females with mental health problems in society because this current finding found mental health issues contribute only a minor role to women’s aggressiveness (21.6%). Hopefully, this finding can become basic research for conducting experimental research in the future related to the same issues of women using a cohort study approach and different instruments, determining the causality between particular mental health problems and aggressive behavior in a diverse sample population because the present study design uses a cross-sectional study that is less likely to detect cause-and-effect relationships than a longitudinal study. Additionally, there are still too few studies in the aggression research literature conducted in the multicultural context of women. Moreover, this current study can become a foundation for other deeper and broader studies in psychology, psychiatry, social sciences, and public health, e.g., the moderating role of the victim mentality in the relationship between mental health and aggression in the female criminal population, and a cross-cultural study related to other predictors of aggression in both men and women.

## 5. Conclusions

In conclusion, this cross-sectional research on mental health and aggression of women, undertaken during the COVID-19 pandemic, states that there is an overall positive relationship between mental health and aggression in women, with a moderate correlation between both variables. Mental health can be a predictor of aggressive behavior, with the influence of mental health on women’s aggression in Indonesia being only 21.6%, which indicates that mental health issues are not a major contributing factor to women’s aggressiveness in society and may help decrease the stigma of women with mental problems, who tend to suffer from more stigma of mental health issues than men, thus increasing the willingness to seek help or treatment. Another finding indicates that differences in age groups (the young adults and middle-year groups) contribute significantly to the relationship between mental health and aggression in Indonesian women. Farther, most of the sample exhibits low aggression levels, at 99.5%. However, nearly half of individuals are affected by mental health problems, at 49.3%. Moreover, a significant proportion, estimated at 25.2%, faces moderate to severe mental health issues. Mental health problems are more common among young adult women at 56.3%, compared to middle-year women at 39.8%.

## Figures and Tables

**Figure 1 behavsci-13-00727-f001:**
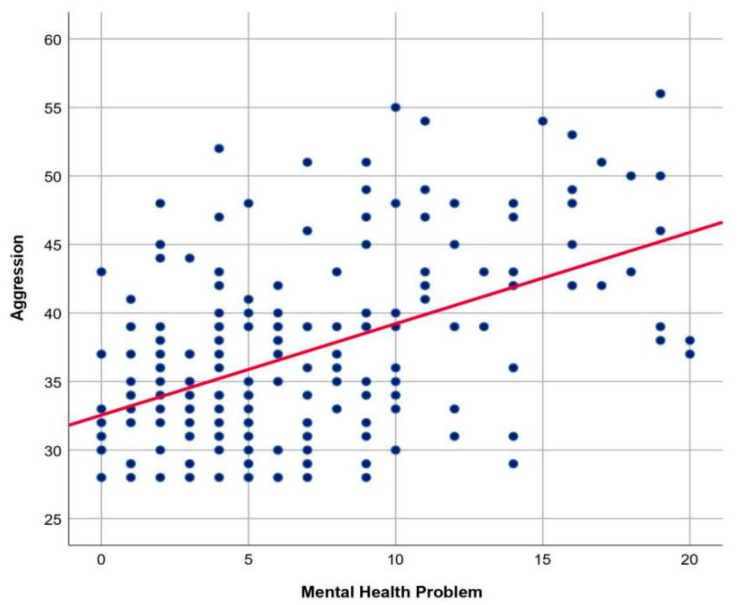
Graph of the relationship between mental health and aggression in Indonesian women. Note: The scatterplot displays a linear positive moderate correlation between mental health problems and aggression.

**Figure 2 behavsci-13-00727-f002:**
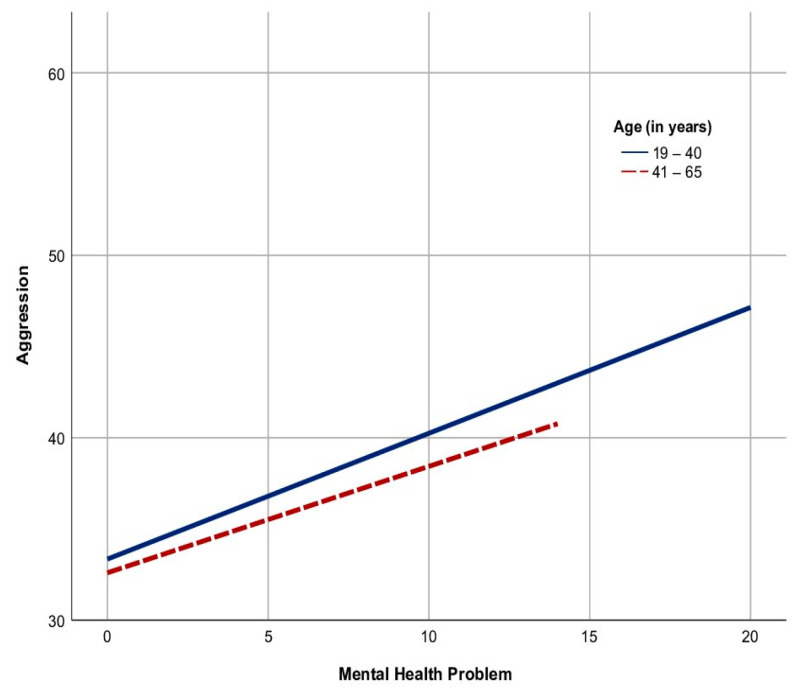
The graph of the relationship between mental health and aggression based on age groups in Indonesian women. Note: The continuous line represents the young adult group, and the dotted line represents the middle-year group.

**Table 1 behavsci-13-00727-t001:** Descriptive analysis summary for mental health predicting aggression in Indonesian women.

Descriptive Statistic Results		N	%	Min	Max	Mean	SD
All ages of participants		203	100	19	67	36.7	12.9
Young adults aged 19–40 years old		119	58.6	19	40	27.4	7.1
Middle years aged 41–65 years old		83	40.9	41	59	49.6	5.4
Later years aged > 65 years old		1	0.5	67	67	67	0
Mental health of all participants		203	100	0	20	6.6	5
Normal condition		103	50.7	0	5	2.7	1.7
Mild problem		49	24.1	6	9	7.4	1.2
Moderate problem		34	16.8	10	14	11.8	1.6
Severe problem		17	8.4	15	20	17.7	1.6
Aggression of all participants		203	100	28	56	37	6.8
Low aggression		202	99.5	28	55	36.9	6.6
Moderate aggression		1	0.5	56	56	56	0
High aggression		0	0	0	0	0	0
Young adults	Normal condition	52	43.7	0	5	3.3	1.6
	Mild problem	27	22.7	6	9	7.4	1.2
	Moderate problem	23	19.3	10	14	11.9	1.7
	Severe problem	17	14.3	15	20	17.7	1.6
	Low aggression	118	99.2	28	54	38.0	6.8
	Moderate aggression	1	0.8	56	56	56.0	0
Middle years	Normal condition	50	60.2	0	5	2.1	1.7
	Mild problem	22	26.5	6	9	7.4	1.3
	Moderate problem	11	13.3	10	14	11.6	1.4
	Severe problem	0	0	0	0	0	0
	Low aggression	83	100	28	55	35.4	6
	Moderate aggression	0	0	0	0	0	0
Later years	Normal condition	1	100	4	4	4	0
	Low aggression	1	100	28	28	28	0

Note. N = number of valid observations for the variables (sample size); % = percentage of samples; Min = minimal value of age or score; Max = maximal value of age or score; Mean = the arithmetic mean across the observations for the participants’ ages or variables scores; SD = standard deviation.

**Table 2 behavsci-13-00727-t002:** Linear regression analysis summary for mental health predicting aggression in Indonesian women.

Statistic Results	r	r-Squared	b	SE	β	t	95%	C.I.
All participants	0.465 ***	0.216	0.231	0.031	0.465	7.451	0.17	0.292
Young adult participants	0.487 ***	0.238	0.271	0.045	0.487	6.038	0.182	0.36
Middle year participants	0.350 **	0.123	0.159	0.047	0.350	3.366	0.065	0.252

Note: ** *p* < 0.01; *** *p* < 0.001.

## Data Availability

The data presented in this study are available on request from the corresponding author. The data are not publicly available due to restrictions on privacy.

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
