# Peer review of "Mental Health and Aggression in Indonesian Women"

_behavsci, 2023, doi:10.3390/bs13090727_

Round 1
Reviewer 1 Report
The article raises an original and interesting idea, which is to delve into how mental health and aggressiveness/violence exercised by women are related, since it usually tends to focus on men.
The article is clear and easily understandable.
Reading the article raises two main concerns for me. First, it raises doubts about the type of study underlying the research. Is it intended to be an epidemiological investigation indicating how women with mental health problems score higher in aggression? In this case, the method should have been in correspondence with the purpose of the study, favoring and taking care of the capture and representativeness of the sample. Furthermore, the fact that only one person presents aggression and that his level of aggression is moderate implies an almost null variability of aggression that invalidates an analysis by categories of the results of aggression as the authors do when they propose different levels of aggression (Table 1). It is possible that other methods and designs would have been more appropriate to know the true relationship between violence and mental health in women. Even the subsequent adoption of a dimensional view establishing the relationship between the scores of the two questionnaires is doubtful, since we are correlating mental health scores in which half of the sample presents mental health problems to some degree with aggressiveness scores in which only one person in the sample presents aggressiveness problems. The results can hardly indicate that mental health affects aggressiveness when there are no people with sufficient levels of aggressiveness in the sample. The second doubt has to do with the instrument used to assess mental health. BSRS is an ultra-brief instrument for which no psychometric information and validity of use for this purpose has been provided. The items that compose it are mental health conditions that are relatively common in daily life (insomnia, being tense, being down) including one item (hostility) that may be relatively redundant with the variable to be related (aggressiveness).
In relation to the above comments, the authors should provide additional information on the BSRS and justify its use for this purpose, as well as provide more information on the method used to capture the sample, its representativeness, its influence on the results and, if appropriate, include in the limitations section what is relevant beyond the sample size.
The question that there is only 0.5% of aggression and that in all cases it is moderate (beyond the fact that it is a lower figure than expected) should be analyzed in the discussion.
Other comments
What are the cut-off points used to make the categories of mental health and level of aggression?
I suppose that the data provided in the conclusions about 95.5% of women with low level of aggression is an error (99.5% in other parts of the text).
Reviewer 2 Report
In the manuscript ,the authors present the useful information about mental health and aggression.The portrayal of Indonesian women is particularly interesting.Please clarify how did the authors obtain informed consent from participants of this Survey. Which test was used to test dana for normality 'distribution?
Discussion:Please discuss the limitations of the study. Discuss the results of this study and compare them with existing knowledge in the literature.Whether the research could be extended to the time after covid .Questionnaire on the socio-economic status of a participant.Please explain which variables were used to describe the socioeconomic status of a participant. Did you used questionnaire that was used in previous surveys or was it constructed for the present research?
Materials and methods: the statistical analysis is adequate.
Reviewer 3 Report
article about Mental health and aggression in Indonesian women
The topics chosen are quite interesting and up to date.
Abstracts are prepared in accordance with scientific principles, which consist of background, methods, results, conclusions and keywords. Suggestions: for abstracts, please discuss the background points directly about existing phenomena, including what kind of aggression the researcher means. Others are appropriate
an introductory chapter that contains background, please emphasize what phenomena you encountered so that it becomes your research topic, of course it is supported by relevant data
the research method is good, what about the inclusion and exclusion criteria? Informed consent and ethical tests have also been included
the results and discussion are quite good, add your opinion to the discussion which is supported by theory
try on the picture or table not to be cut off between the title and the table / picture
conclusions must answer the research objectives
references are used accordingly
Round 2
Reviewer 1 Report
The authors have conveniently addressed most of the suggestions and comments. The changes and added information are useful and improve the article. Nevertheless, the main indication and doubt about the value of the research has not been addressed, probably because it cannot be. The aim is to establish the relationship between mental health and aggressiveness in women, although only one woman out of 236 presents a significant level of aggression (also moderate-low). Therefore, the results would indicate the relationship not between mental health and aggressiveness in general terms, but in a population of low aggressiveness. This is easily observed in Figure 1 which includes a scatterplot showing a linear positive moderate correlation between mental health problems and aggression. . Therefore, the regression analysis indicating an influence of mental health on the aggression of women (21.6% variance explained) would be for people with low levels of aggression since the sample does not include people with moderate or high levels of aggression. The variance could vary significantly in a different sample. In summary, if we take the sample selection as good, we should conclude that Indonesian women have low levels of aggression and that should be an important conclusion of the study that has not been highlighted.
Round 3
Reviewer 1 Report
no new comments